# Development and Validation of a Box and Flux Model to Describe Major, Trace and Potentially Toxic Elements (PTEs) in Scottish Soils

**DOI:** 10.3390/ijerph18178930

**Published:** 2021-08-25

**Authors:** Luigi Gallini, Andrew Hursthouse, Antonio Scopa

**Affiliations:** 1Liceo Scientifico “Carlo Cattaneo”, Dipartimento di Scienze, Via Sostegno, 41/10, 10146 Torino, Italy; 2Sezione di Chimica Agraria, Di.Va.P.R.A., Università di Torino, Largo Paolo Braccini, 2, 10095 Grugliasco, Italy; 3School of Computing, Engineering & Physical Sciences, University of the West of Scotland, Paisley PA1 2BE, UK; andrew.hursthouse@uws.ac.uk; 4Scuola di Scienze Agrarie, Forestali, Alimentari ed Ambientali (SAFE), Università della Basilicata, Vialedell’Ateneo Lucano, 10, 85100 Potenza, Italy

**Keywords:** soil property variability, soil contamination, pseudo-total elemental pool, box and flux mathematical model

## Abstract

The box and flux model is a mathematical tool used to describe and forecast the major and trace elements perturbations of the Earth biogeochemical cycles. This mathematical tool describes the biogeochemical cycles, using kinetics of first, second and even third order. The theory and history of the box and flux modeling are shortly revised and discussed within the framework of Jim Lovelok’s Gaia theory. The objectives of the investigation were to evaluate the natural versus anthropic load of Potentially Toxic Elements (PTEs) of the Scottish soils, investigate the soil components adsorbing and retaining the PTEs in non-mobile species, evaluate the aging factor of the anthropic PTEs and develop a model which describes the leaching of PTEs in layered soils. In the Scottish land, the soil-to-rock enrichment factorinversely correlates with the boiling point of the PTEs. The same is observed in NW Italy and USA soils, suggesting the common source of the PTEs. The residence time in soils of the measured PTEs linearly correlates with the Soil Organic Matter (SOM). The element property which mostly explains the adsorption capacity for PTEs’ is the ionic potential (IP). The downward migration rates of the PTEs inversely correlate with SOM, and in Scottish soil, they range from 0.5 to 2.0 cm·year^−1^. Organic Bentoniteis the most important soil phase adsorbing cation bivalent PTEs. The self-remediation time of the polluted soil examined ranged from 50 to 100 years. The aging factor, the adsorption of PTEs’ into non-mobile species, and occlusion into the soil mineral lattice was not effective. The box and flux model developed, tested and validatedhere does not describe the leaching of PTEs following the typical Gaussian shape distribution of the physical diffusion models. Indeed, the mathematical model proposed is sensitive to the inhomogeneity of the layered soils.

## 1. Introduction

In the geodynamic earth system, elements cycle between several compartments, most relevant of which are the atmosphere, hydrosphere, pedosphere, biosphere and lithosphere, but also across ecosystems in biogeochemical equilibria with these principal planetary reservoirs. As demonstrated by Jim Lovelock’s Gaia theory, many of the important driving forces of the earth’s dynamic biogeochemical system, as well as most of the relevant geochemical reaction kinetics, rule the flux of elements between the different earth reservoirs, modulated by the biological activity of living organism [1,2]. Most of the chemical equilibria controlling geodynamics are modulated by simple and multiple retroactive feedback, which ensures some buffering capacity to earth ecosystem and some degree of homeostasis to overall planet earth [3].

Any chemical element is characterized by specific concentration ranges within which the elements may affect organisms living in an ecosystem. The range of Potential Toxic Elements (PTEs) include those with direct toxicity or where they exceed the range of concentration essential or needed for the ecosystem stability and development [4].

Soil as part of the dynamic earth system is the most important planetary sink for many of the PTEs released in the environment by human activity, and since the beginning of the industrial revolution in the 18th Century, there has been an exponential increase in most of the earth environmental ecosystems which are in biogeochemical equilibria, and in particular with soils and sediments.

The intensive agricultural and industrial activity has led to PTE accumulation in agricultural soils threatening the environment, as well as the quality, quantity and security of cultivated products [5]. Many anthropogenic PTEs flushed from contaminated soil into the aquatic ecosystem—and consequently, negatively affecting its biodiversity and stability—eventually accumulate in the marine food chain and affect the security of fish, and the state of human health and fertility rates [6].

The biogeochemical cycles of major chemical elements among earth compartments can be controlled by first-, second-, or even third-order kinetics, but generally the biogeochemical cycles of the PTEs in trace amounts can be described by simple first-order kinetics. The biogeochemical cycles of individual major and traces elements can be coupled together to achieve more detailed and complex models carefully describing the biogeochemical equilibria [7]. It is reasonable to believe that, by coupling together the major element biogeochemical cycles with trace element cycles, we can achieve a more detailed Earth Geochemical Reference Model (GERM). This geochemical reference model could find wide application in the environmental impact assessment of contaminated land and ecosystems (see https://earthref.org/GERMRD/ (accessed on 21 April 2021)).

To evaluate the anthropogenic contribution versus natural origin of the PTEs detected in the environmental compartments, it is critical to estimate the chemical kinetics of the interactions of PTEs with minerals, rocks, solution, gasses and interaction of the bioavailable PTEs with the organism living in the relevant environmental compartment. Fate, mobility, distribution and bioavailability of PTEs depending on the chemical–physical properties of soils, rocks and sediments, which are in turn determined by chemical–physical kinetics reactions mostly controlled by living organisms (these include: complexation, hydrolysis, dissolution, solubility, oxidation, reduction, precipitation, adsorption, methylation and release into atmosphere or translocation, and leaching toward groundwater, uptake into biota, uptake into terrestrial biomass and the related ecosystems) [8]. Most of the rates of the chemical–physical reaction kinetics reported in the literature are enhanced by enzymatic system of living organisms by several orders of magnitude greater than the basic inorganic kinetic reaction rates. As a general rule, the PTEs persist in the earth ecosystem due to the long time required to isolate them in poorly-mobile and non-bioavailable chemical species [9,10].

Box and flux models have been employed to describe the fate and behavior of anthropogenic PTEs fluxes in localized ecosystems. Historically, the box and flux model was initially developed to describe and forecast the leaching of radionuclide in soils [11,12]. With regionalized and georeferenced box and flux models used to describe and forecast the fate and the effects of anthropogenic fluxes of major and trace PTEs in ecosystem Asian oceanic waters [13,14]. The same approach was used to describe and forecast the fluxes of anthropogenic PTEs in the Canadian continental agricultural ecosystem [13,15,16,17,18,19], and employed to describe trace PTEs leaching from soils to groundwater in very localized cases [12,15,20]. Ultimately box and flux models were used to describe and forecast the fluxes of PTEs contaminating pedosphere into the food chain and to evaluate the anthropogenic perturbation of biogeochemical cycles.

Therefore, box and flux models are mathematical tools that are widely used to not only model the earth biogeochemical equilibria, but also the response to perturbation of these equilibria. The box and flux model have been applied to describe the major elements C, N, S and P biogeochemical cycles, but also to evaluate the short- and long-term effects caused by industrial growth that are kinetically perturbing the biogeochemical cycles of the major elements and trace PTEs present in the environmental ecosystem.

In this study, we used local soil systems to determine the soil–rock enrichment factor of PTEs, the anthropogenic PTE load and the properties of the soil that affect the mobility of that pollution. Furthermore, we evaluate the chemical–physical processes and kinetics that remove low-mobile anthropogenic PTEs adsorbed on soil colloids, in insoluble amorphous precipitates and in non-bioavailable phases occluded in the soil, forming crystalline mineral lattices. The ultimate aim of these measures was to develop and validate a new box and flow model that describes the mobility of PTEs in soils and, finally, to determine the so-called “self-remediation” or flushing rates of anthropogenic soil PTEs.

## 2. Materials and Methods

### 2.1. Description of Study Area

The soil selected to test and validate the model was developed on the top of a drumlin, several hundred meters long, located in Clelland, South Glasgow, Scotland, UK. The quaternary geology map of the British Geological Survey Service (www.largeimages.bgs.ac.uk/iip/mapsportal.html?id=1003899 (accessed on 21 April 2021)) indicated the soil being studied was derived from a fluvial–glacial parent material that had been deposited during the last glaciations. The alluvial soil was classified as Gleyic Fluvisol [21]. It has been polluted by atmospheric emissions from an adjacent smelter operation, peaking in the 1950s until 1987 when it was decommissioned. Climatic conditions of the area, average of the monthly data, was collected by the Scottish meteorological service (www.metoffice.gov.uk (accessed on 11 June 2000, stored by Gallini L.)), from 1990 to 1995, and are shown in Figure 1.

### 2.2. Soil Profile

Ap horizon (0–25 cm). This was a typically eluvial ploughed horizon, with a sharp boundary and a strong brown color due to the organic matter content. The porosity was high, and the plasticity moderate. The soil aggregates were irregular. Throughout the horizon, strong biological activity was observed: worms and abundant grassroots were found. A relevant flux of PTEs from soil to land biosphere could be expected from Bt horizon due to the intense plants root activity and plants uptake.

Ae horizon (25–35 cm): This was an eluvial horizon below the plough layer. It had a more diffuse boundary with respect to the Ap horizon and a yellow-brown color, which indicates the presence of Goethite and may be the rare mineral forming Maghemite, recognized as an indicator of biomineralization. These iron oxides are shown in the literature to be potential biominerals whose crystallization process is mediated by soil bacterial species. The soil-layer colors and presence of the yellow Goethite indicate a poor drainage state and suggest a seasonal redox oscillation within the Ae horizon, which is confirmed by the estimated soil porosity and the hydrologic balance between precipitation and evapotranspiration (EPT). The Ae horizon had low plasticity and moderate porosity. The soil aggregates were polyhedral with irregular fractures. The cracks observed in the Ae horizon suggest periodic changes in water saturation of the soil and the presence in it of swelling clay minerals in some uncertain amount but below 5%. The biological activity was low and reasonably was mainly represented by soil microflora and soil microfauna.

Bt horizon (35–40 cm). It was an illuvial horizon containing yellowish and greenish lenses that suggested patches of strongly reducing conditions. The red color indicated the presence of iron oxides, such as Hematite, and may be inherited Ilmenite. Porosity and plasticity of the horizon were very high. Soil aggregates were polyhedral with regular fractures, a feature that is proving that swelling clay minerals are present in the Bt horizon in more amount than the upper one Ae. Thin clay films were observed in some polyhedral cracks between aggregates, suggesting that translocation of clay minerals from the upper soil horizon is effective, and they accumulate in the Bt horizon in some quantity. In the Bt horizon, we observed cm diameter stones containing strongly weathered lumps ofcoal and mixtures of basaltic and sedimentary rock.

### 2.3. Soil Sampling

Superficial debris and vegetative materials were removed before the collection of samples from the soil surface. The soil was sampled by collecting slices of 5 cm thick at 5 cm increments from a surface of 20 × 20 cm^2^ (about 3 kg dry weight for each layer). Soils were collected in plastic bags, homogenized and air-dried to constant weight, at room temperature; they were then softly disaggregated and sieved to remove coarse debris and rubble (>2.0 mm).

### 2.4. Analytical Methods

Physical-chemical parameters of soil were determinate in accord with standard methods [22]. The exchangeable pool was determined by extraction of 2 g soil samples (dry weight equivalent) suspended in 20 mL of BaCl_2_ (10% w/v at pH = 5.5), so that the precipitation of metal oxides during extraction and the dissolution of any mineral phases were avoided, and were mechanically shaken for 30 min and sonicated for 15 min. The extraction procedure was applied five times until the residue of the extractable pool was negligible. The supernatant was collected by centrifugation and analyzed by ICP–AES or HGA–AAS.

Concentrations of “pseudo-total” metals were determined after digestion of soil samples with aqua-regia, which does not completely destroy silicates [21,23]. Then 2.0 g of dry soil was placed in a digestion vessel with 6 mL of HCl and 14 mL of HNO_3_ and digested in microwave digestion unit (MLS-1200 Mega, Milestone Inc., Shelton, CT, USA), at rising temperature steps, and later filtered. The digested samples were analyzed for Si, Al, Fe, Ca, Mg, K, Na, Ni, Co, Cu, Zn, Cr, Mn, P and S by inductively coupled plasma–atomic emission spectrometry (ICP–OES; model iCAP 6000, Thermo-Scientific, Cambridge, UK). A certified standard soil reference material was also analyzed in order to evaluate the method of digestion and the accuracy of analysis (GBW7404). The external standard solutions were derived from 1000 ppm stock metal solution; multi-element standards for calibration were prepared in the same reagents to minimize interferences and matrix effects. Three replicate analyses for samples were performed for all determinations. The Standard Deviation (SD) was below 10% for pH, texture, Cation Exchange Capacity (CEC), and below 1% for the aqua-regia extractable pool [16,23].

All the chemicals used in this study were analytical grade or better (Sigma-Aldrich, St. Louis, MO, USA) and used without further treatment. Reagent solutions were prepared with deionized water of resistivity not less than 18.2 MΩ×cm^–1^. The main soil properties are reported in Table 1.

### 2.5. Modification of the Bonazzola’s Model 

The box and flux model here developed and tested isthe generalization of the model of Bonazzola et al. [12]. The main differences of the new box and flux model are the following:(a)In the top surface reservoirs, a constant atmospheric input I(t) can be considered.(b)In the model here proposed, each soil reservoir (S_n_) is characterized by a specific kinetic constant (K_n_) which characterizes any different soil layer—indeed, the averaged K kinetic constant fitted across the entire soil profile, such as the Bonazzola’s box and flux model calculates.(c)The general solution valid for Sn reservoir is found and proposed—indeed, the only four reservoirs model solution proposed by Bonazzola et al. [12].

As in the Bonazzola’s model, the concentration of the PTEs in the soil profile are subsided in reservoir “Sn”, and the leaching from a reservoir to the under layer one is assumed to be a first order kinetics. The leaching of the PTEs can be so described by the system of n homogenous differential equations listed below.
dS_0_/dt = I − F_0_ = I − (K_0_ × S_0_)
dS_1_/dt = F_0_ − F_1_ = (K_0_ × S_0_) − (K_1_ × S_1_)
dS_2_/dt = F_1_ − F_2_ = (K_1_ × S_1_) − (K_2_ × S_2_)
dS_3_/dt = F_2_ − F_3_ = (K_2_ × S_2_) − (K_3_ × S_3_)
dS_n−1_/dt = F_n−2_ − F_n−1_ = (K_n−2_ × S_n−2_) − (K_n−1_ × S_n−1_)
dS_n_/dt = F_n−1_ − F_n_ = (K_n−1_ × S_n−1_) − (K_n_ × S_n_)

The system is resolvable, and the general solutions are as follows:(1)S_(1,t)_ = I/K_(1)_ − (I/K_(1)_ × S_(1,0)_)exp(−K_(1)_ × t)(2)..........(n)S_(n,t)_ = I/K_(n)_+ [∑(m = 1→m = n) (K_(n−1)_/(K_(n)_− K_(m)_) × A_(m,n−1)_)exp(−K_(m)_ × t)] − [I/K_(n)_S_(n,0)_ + ∑(m = 1→m = n) (K_(n−1)_/(K_(n)_ − K_(m)_) × A_(m,n−1)_]exp(−K_(n)_ × t)]where

n > 1

A_(1,1)_ = (S_(1,0)_ − I/K_(1)_)

A_(n,m)_ = [S_(n,0)_ − I/K_(n)_ − ∑(m = 1→m = n)(K_(n−1)_/(K_(n)_ − K_(m)_) × a_(m,n−1)_]exp(−K_(n)_ × t)]

Three geochemical parameters can be derived from the kinetic constant (K_n_) and calculated for each soil reservoir (S_n_): (a) residence time of the pollutants in the reservoir T_r_; (b) half-life time of the pollutant in the reservoir, t_½_; and (c) average downward migration rate V_n_ of the pollutant in the reservoir S_n_. The relative formulas are listed as follows:Tr_n_ = K_n_^−1^
t_½n_ = ln2 × K_n_^−1^
V_n_ = K_n_ × d_n_,
where d_n_ is the thickness of the reservoir S_n_.

The residence time (Tr_n_) represents the ratio between the amount of the element in the reservoir and the flux of element throughout the reservoir and at the steady state of the system; the residence time represents the mean time of an element that is remaining in the reservoir.

The half-life time (t_½n_) at the steady state of the system is the time required to remove half of the element pool in the reservoir n. The lower the half-life, the faster the turnover of the element in the reservoir.

The downward migration rate (V_n_) represents the average downward migration rate in the reservoir (S_n_), and it is related to the partition constant for the adsorbing phases following the equation:
V_n_ = V_0_ × exp(−Σ_1_^n^Kd_i_ × Q_i_)
where V_0_ is the downward migration rate in absence of the adsorbing phase, and Kd_i_ and Q_i_ are the partition coefficient and the pseudo-total amount of the adsorbing phases “i” respectively pooled in the reservoir S_n_. The downward migration rate in absence of adsorbing phases can be assumed to be as follows:
V_0_ = (P × I_c_ − ETP)/θ
where P is the rainfall rate, I_c_ is the infiltration coefficient, ETP is the sum of EPT and θ is the soil porosity.

When the values I_(t)_, S_(n,0)_ and S_(n,t)_ are known the values of the kinetic constant Kn characteristic of the reservoir Sn can be numerically calculated. To calculate K_n_ from the value of I_(t)_, S_(n,0)_ and S_(n,t)_, the solver extension for Excel was used, and the validity of the solutions found were checked by a self-produced software.

Previous studies have discussed whether the box and flux models applied to forecast the leaching of chemicals in soils are mathematical rather than physical models. Some authors agree that the mathematical box and flux models are predominantly equivalent to the advective–diffusion model at an infinite scale, e.g., when the S_(n,t)_ reservoirs are infinite. A truly physical advective–diffusive model applied to soils requires the determination of several soil properties, such as infiltration rates, soil porosity, pore distribution, soil tortuosity, exchange sites selectivity distribution, colloids surface charge and many other soil properties difficult to be measured, detected or estimated. In this study, a box and flux mathematical model was developed and applied instead of a truly advective–diffusive physical model, as it is more straight forward to develop and numerically solve, and it useful when applied to a layered substrate, such as soil profile [24].

### 2.6. Hypothesis Applied in the Box and Flux Model and Validation

The history of pollution inputs into the soil is known; the values I_(t)_ and S_(n,0)_ were estimated by the value S_(n,t)_ of the trace elements. Since the beginning of the 1900s until 1987, the soil was ploughed, and therefore the atmospheric input I_(t)_ mostly accumulated in the plough layer [16]. After 1987, the industrial plant was closed and subsequently the ploughing of the soil ceased and cropping for barley was changed to grassland. Therefore, the trace element atmospheric input I_(t)_ reduced close to zero, and the elements in the ploughed layer was dominated by downward leaching. The input of the trace elements by agricultural fertilizer are assumed to be negligible with respect to the atmospheric input I_(t)_ following Riffaldi and Levi-Minzi [25].

To calculate the kinetic constant (K_n_), the values of I_(t)_, S_(n,0)_ and S_(n,t)_ must be identified and were estimated by the following assumptions:(a)From the beginning of the 1900 until 1987, the soil received an atmospheric input I_(t)_ which has appreciably increased the trace elements concentration in soil;(b)The atmospheric input I_(t)_ can be disregarded after the end of smelter activity;(c)Until 1987 the soil was ploughed, and therefore the I_(t)_ input accumulates in the ploughed layer and the leaching of trace elements is minimal, below 5% of the measured anthropogenic PTEs load;(d)Almost all trace elements were held in the plough layer until the end of ploughing;(e)Trace elements starts to be leached throughout the layers of the soil profile after the end of ploughing;(f)The biological uptake and biological loop of soil-plant-soil transferring fluxes is minimal and can be disregarded at the present time;(g)The lithogenic contribution to the trace element pool extracted by aqua-regia can be disregarded, and the S_(n,t)_ value of PTEs can be estimated by the aqua-regia extraction at the time of sampling;(h)Most of the trace elements pool was bound to the labile fraction and the ageing factor is reduced. Therefore, the value of S_(n,t)_ can be estimated by the aqua-regia extraction at the time of sampling. Anthropogenic PTEs are bound or immobilized in the soil biomass, associated with humic and fulvic compounds, labile soil mineral fraction and biominerals. Anthropogenic PTEs estimated by aqua-regia is almost exhaustive, and only a partially overestimated due to partial dissolution of primary mica by aqua-regia extraction [23,24].

### 2.7. Statistical Analysis

SPSS statistical package (Window version 25, SPSS Inc., Chicago, IL, USA) was used for data analysis. The analysis of the experimental data was carried out by using one-way ANOVA, Duncan multiple comparison test and Pearson correlation analysis. All significance statements reported in this study are at the *p* < 0.05 level [26,27].

## 3. Results and Discussion

### 3.1. Soil Profile Properties

The very coarse fraction reflects the sedimentary (primary) features of the parent materials because they are unaffected by pedological processes [28]. The stones are scattered in the Ap horizon, caused by agricultural disturbance, and progressively decrease to zero towards the bottom layer. It is suspected that the parental material was originally sediment in shallow and slow-moving water in fluvial–glacial environment, affected by current activity. A portion of the clay in the Bt horizon is likely to be primary and due to the Würm post glacial warm period. The pedological horizons reflect the soil clay content, which is homogeneous in throughout the three detected horizons. Only a slight increment in clay content is observed in the top layer of the Ap horizon and in the bottom layer of the Ae horizon. These variations are common in soils with high vertical flushing by soil water and assisted by organic colloids from the surface layers. This is a typical condition in a boreal Atlantic strong-weathered soil environment due to the high rainfall rates. The Soil Organic Matter (SOM) content is highest in the Ap horizon and exponentially decreases sharply in the Ae horizon. There is an increase in the organic matter towards the top of the Ap horizon because of the root zone and degrading organic matter accumulating in the soil canopy. The CEC increases both toward the top and the bottom of the profile. Both organic and inorganic colloids appear to affect the CEC and strong multiple correlations are observed between CEC, SOM and clay content: CEC = (0.144 ± 0.031) × Clay + (0.829 ± 0.185) × SOM; n = 9, r^2^ = 0.948; F = 9.1 × 10^−5^

The clay and organic matter contribution to CEC can be estimated respectively as 14 and 80 Eq·kg^−1^. The clay minerals have a low CEC and poorly crystalline illitic clay minerals are expected to dominatein the soil profile. The amount of exchangeable Na, K, Mg and Ca (Eq·kg^−1^) exceeds the CEC; therefore, the soil has an 100% base saturation and is oversaturated with respect to carbonates, condition which theoretically stabilize the formation of pedogenic swelling clay minerals.

The pH ranges from sub-acid to acid and is quite constant along the soil profile. A little increase of the pH is observed in the organic top layer, and a subtle decrement is observed in the clay minerals rich bottom layer of the Bt horizon. The observed pH range is above the pK both of the SOM carboxyl and the clay minerals basal surfaces but is below the pK of Mn, Fe, and Al oxides surfaces. The SOM and the clay minerals are expected to be mostly negatively charged and the oxides are positively charged in the pH range observed in the soil profile. The exchangeable H^+^ was estimated by pH_(KCl)_ and was constant through the Ap horizon, and increases exponentially below it, in agreement with the observations on the soil colloids, and CEC.

Reducing conditions are evident in the Bt horizon where gray-greenish lens is observed. The relative humidity (RH) can be considered as a broad indication of reducing conditions and correlate with the clay and SOM:RH% = (6.311 ± 2.943) + (0.277 ± 0.060) × Clay + (1.993 ± 0.331) × SOM; n = 9, r^2^ = 0.884, F = 1.58 × 10^−3^

A reducing environmental low oxidative potential is expected to prevail in the bottom of Bt horizon due the high quantity of clay, high relative humidity and low oxygen activity. The pH value is sub-acid for most of the Ap and Ae horizons, but it results in being acid in the Bt, where it is probably buffered by the dissolution and precipitation reactions of oxides.

### 3.2. Pseudo-Total Elemental Pool

The aqua-regia extracts all the PTEs bound to the soil humic and fulvic compounds, all the PTEs incorporate into soil microfauna and microflora pool, almost all the PTEs bind to soil minerals labile pool, most of the PTEs bound to superior vegetal residues and only a little fraction of the PTEs sinks into primary soil forming minerals. Most of the elements extracted by aqua-regia are therefore the anthropogenic soil PTEs concentration, here mainly due to the smelter plant emission in the atmosphere and its deposition on the top investigated soil layers. The concentration of pseudo-total element pool extracted in the investigated soil reservoirs is reported in Table 2.

It is evident from Table 2 that the PTEs load throughout the examined soil profile is inhomogeneous among the different soil layers, and specific for any element. No element studied shows a Gaussian distribution across the profile, as usually observed in most of the soil analysis published in the literature and observed in the field [23]. It is hypothesized that the PTEs load in any soil layers reflects the adsorbed component in the soil, as well as specific chemical properties of individual PTEs. 

When comparing the aqua-regia extracted pool of the trace elements with the averaged soil composition reported in the literature, it is apparent that most of the trace elements in the investigated soil have a concentration closed to the global soil average [29,30]. Only in the case of Cd is the soil averaged value exceeded by the aqua-regia’ extract soil-to-crust enrichment factor, which is close to two. It should be noted that the trace metal concentration in the aqua-regia extractable pools and the soil average concentration reported in the literature are not directly comparable, because the global average soil composition quoted in the literature results from a total (HF) digestion and not by aqua-regia extraction. The aqua-regia soil extract and does not allow us to estimate the PTEs occluded in most of the silica primary minerals, and it is believed to be almost everywhere strongly sensitive to anthropogenic soil PTEs load [23,25].

The correlation matrix among the elements of the aqua-regia pool is shown in Table 3, and it allowsus to broadly distinguish four groups of the elements with similar distribution in the soil profile. The groups are (a) Na; (b) Al, Fe, Mg, K and Si; (c) Ni, Co, Cu, Zn, Cd, Pb, Mn and Ca; and (d) Cr, P and S.

The concentration of Na has a complex distribution throughout the soil profile and its pseudo-total concentration does not correlate with any other elements. It is reasonable to suppose that the concentration of Na in the soil is affected by the sea aerosol input for the sample site from on-land marine transfer. The concentration of K, Fe, Mg, Al and Si was almost constant within the soil profile, and a concentration peak of these elements are observed in the Bt horizon. Elements K, Mg, Al, Fe and Si results strongly correlate to the clay fraction content, and therefore is reasonable believe that the soil clay minerals are the main source of the aqua-regia extracted pool of these soil chemical element. The group of the bivalent cation elements Ni, Co, Cu, Zn, Cd, Pb, Mn and Ca has complex patterns in the distribution of the concentration along soil profile. Correlations among the concentration of these elements in the soil profile are observed, and correlation varies from weak to strong. The concentrations of the elements of this group are correlated with SOM and for some elements even with the soil clay fraction, such as Ca, Co and Ni. The group of elements stable as anions (Cr, P and S) has a pseudo-total concentration almost monotonically decreasing across the soil profile. Positive strong correlations between pseudo-total pool and organic matter are observed, and it is reasonable to assume that these anionic species are strongly bounds to the SOM colloids.

### 3.3. Soil Enrichment Factor

The soil to crust enrichment factor is the ratio between the element concentration in the soil and the element concentration in the crust. The soil enrichment factor (SEF) in the investigated soil was calculated by assuming that the pseudo-total pool of the measured elements is due to the anthropicfluxes and to the perturbation of the natural biogeochemical cycles, dividing the aqua-regia extracted pools values for the average crust concentration of the PTEs. The SEF calculated for the soil is reported in Table 4.

The enrichment factor for Ca, Mn, Cd, Co, Cr, Cu, Ni, Pb, Zn, P and S in the studied soil result correlates with the USA soil-to-crust enrichment factor. 

If the SEF of the S is disregarded, a strong correlation is observed between the pseudo-total SEF in Clelland and the soil enrichment in the USA:SEF *Clelland* = −(1.80 ± 0.43) + (4.71 ± 0.35) × SEF*_USA_*; n = 9, r^2^ = 0.963, F = 2.83 × 10^−6^

This result is in agreement with previous investigation on anthropic Zn, Pb and Cd in NWItaly [30]. The SEF of Cd, Co, Cr, Cu, Mn, Ni, Pb and Zn observed in the soil does not correlate with the ionic potential (IP) of the element; instead, it negatively correlates with the boiling temperature of the element (Figure 2). The correlation observed between SEF and element boiling temperature suggests that the SEF of the PTEs is due to the smelter plant activity rather than soil chemical pedological processes both in Clelland, NW Italy, the USA and in Canadian soils [13,15,16,18,31].

### 3.4. Residence Time of Element in the Reservoirs

The residence times of the pollutants in the reservoirs of the soil profile are shown in Table 5.

The average residence time (year) is in the following order:Na_(3.3)_ < Ca_(4.8)_ < Co_(5.5)_ ≈ Si_(5.7)_ < Ni_(6.3)_ ≈ Cu_(6.2)_ < Zn_(7.5)_ < P_(7.8)_ < S_(10.1)_ < Cd_(11.7)_ < Pb_(23.9)_ < Mn_(28.9)_

The residence time in the top soil profile is maximum, decreases downward following organic matter distribution and increases again in proximity to the Bt and beyond.

The soil properties which control the residence time have been investigated by forward multiple regression analysis. The main soil property which controls the residence time of Na is the clay content. The main soil property which controls the residence time of the bivalent cations, i.e., Ca, Co, Ni, Cu, Zn, Cd, Pb and Mn, is the SOM, while for the anions (Si, P and S), it is both the organic matter and clay fraction:RT(Na) = (−2.826 ± 1.580) + (0.251 ± 0.630) × Clay; n = 8, r^2^ = 0.724, F = 1.52 × 10^−4^
RT(Cu) = 0320 (±1469) + 1.09 (±0.24) × SOM; n = 8, r^2^ = 0776, F = 3.83 × 10^−3^
RT(Co) = −0.686 (±0.939) + 1.145 (±0.153) × SOM; n = 8, r^2^ = 0.904, F = 2.85 × 10^−4^
RT(Zn) = 0.340 (±1.953) + 1.349 (±0.318) × SOM; n = 8, r^2^ = 0.750, F = 5.40 × 10^−3^
RT(Ca) = −1.50 (±1.10) + 1.422 (±0.178) × SOM; n = 8, r^2^ = 0.914, F = 2.06 × 10^−4^
RT(Ni) = −2.68 (±1.73) + 1.68 (±0.28) × SOM; n = 8, r^2^ = 0.857, F = 9.78 × 10^−4^
RT(Cd) = −6.361 (±5.098) + 3.37 (±0.83) × SOM; n = 8, r^2^ = 0.734, F = 6.59 × 10^−3^
RT(Mn) = 5.40 (±7.22) + 4.40 (±1.17) × SOM; n = 8, r^2^ = 0.700, F = 9.58 × 10^−3^
RT(Pb) = −13.3 (±4.22) + 6.95 (±0.67) × SOM; n = 8, r^2^ = 0.945, F = 5.38 × 10^−5^
RT(Si) = −9.65 (±2.80) + 0.360 (±0.114) × Clay + 1.210 (±0.179) × SOM; n = 8. r^2^ = 0.933, F = 1.17 × 10^−3^
RT(S) = −17.80 (±3.97) + 0.568 (±0.162) × Clay + 2.597 (±0.254) × SOM; n = 8, r^2^ = 0.966, F = 2.15 × 10^−4^
RT(P) = −12.70 (±3.23) + 0.505 (±0.132) × Clay + 1.517 (±0.207) × SOM; n = 8, r^2^ = 0.945, F = 7.02 × 10^−4^

Assuming that the SOM is the dominant soil property controllingresidence time (RTE) of the elements in the topsoil layer, the RTE can be described by the following simple linear model:RTE = α × SOM

The parameter α has been calculated by multiple regression for Ca, Co, Ni, Cu, Zn, Cd, Pb, Mn, Si, P and S; the results are shown in Table 6, and the slopes of the linear regression results α has been correlated withthe IP of the element.

The results differ between divalent cations and anions. For the divalent cations, the constant α is inversely proportional to the IP of the metal, while, for the anions, it is directly proportional to the IP (Figure 3 and Figure 4).

The regressions observed between parameter α and the IP of the elements are as follows:Bivalent cations: α = (7.85 ± 1.95) − (2.50 ± 0.80) × IP − 1; n = 8, r^2^ = 0.622, F = 2.00 × 10^−2^
Anions: α = (1.88 ± 0.58) + 0.306 (±0.05) × IP; n = 3, r^2^ = 0.973, F = 1.05 × 10^−1^

### 3.5. The Downward Migration Rate of the Pollutants

The calculated downward migration rates are listed in Table 7. The downward migration rates increase in the following order:S_(0.89)_ < Zn_(0.92)_ = P_(0.92)_ ≈ Pb_(0.93)_ < Mn_(1.06)_ < Cu_(1.14)_ < Ca_(1.21)_ ≈ Ni_(1.23)_ ≈ Co_(1.24)_ ≈ Si_(1.26)_ < Cd_(1.42)_ < Na_(1.75)_

As a general rule in the profile, the downward migration rates have a minimum in the top layer, increase to a maximum in the Ae horizon and decrease again approaching the Bt horizon.

It has been assumed that the abundance of the adsorbing phases should be linearly correlated with the logarithm of the downward migration rates, V_(E)_, of the element. This has been tested by multiple regression analysis. In Table 8, the correlation matrix between ln(V_E_) and the soil properties is shown. Most of the PTEs correlate both with SOM and CEC, but the correlation with clay content alone is poor.

All the soil properties correlated to the lnV(_E_) have been investigated by using the forward multiple regression method. The meaningful multiple regression funds are listed below.

ln(V_Na_) = (2.12 ± 0.53) − (0.0663 ± 0.0210) × Clay%; n = 8, r^2^ = 0.624, F = 1.96 × 10^−2^

ln(V_Co_) = (1.033 ± 0.101) − (0.177 ± 0.016) × SOM; n = 8, r^2^ = 0.951, F = 3.81 × 10^−5^

ln(V_Ni_) = (1.164 ± 0.130) − (0.211 ± 0.021) × SOM; n = 8, r^2^ = 0.944, F = 5.69 × 10^−5^

ln(V_Cu_) = (0.824 ± 0.278) − (0.160 ± 0.045) × SOM; n = 8, r^2^ = 0.676, F = 1.22 × 10^−2^

ln(V_Zn_) = (0.625 ± 0.265) − (0.160 ± 0.043) × SOM; n = 8, r^2^ = 0.698, F = 9.76 × 10^−3^

ln(V_Cd_) = (1.259 ± 0.637) − (0.281 ± 0.104) × SOM; n = 8, r^2^ = 0.550, F = 3.48 × 10^−2^

ln(V_Pb_) = (1.018 ± 0.571) − (0.365 ± 0.093) × SOM; n = 8, r^2^ = 0.720, F = 7.76 × 10^−3^

ln (V_Mn_) = (0.615 ± 0.224) − (0.120 ± 0.036) × SOM; n = 8, r^2^ = 0.645, F = 1.64 × 10^−2^

ln (V_Ca_) = (1.093 ± 0.130) − (0.201 ± 0.021) × SOM; n = 8, r^2^ = 0.938; F = 7.73 × 10^−5^

ln(V_S_) = (0.911 ± 0.223) − (0.240 ± 0.036) × SOM; n = 8, r^2^ = 0.880, F = 5.73 × 10^−4^

ln (V_P_) = (0.687 ± 0.252) − (0.173 ± 0.041) × SOM; n = 8, r^2^ = 0.748; F = 5.53 × 10^−3^

The ln(V_Na_) correlates with clay content only. For all bivalent cations and all anions, the ln(V_E_) the result correlates linearly with SOM (Table 9).

Forward multiple regression analysis demonstrates that the bulk clay fraction is not meaningfully correlated to ln(V_E_) when SOM and clay fraction are considered together. It is reasonable to assume that the downward migration rate of the considered PTEs is sensitive of the mineralogical composition to the clay fraction, as previously demonstrated for ^137^Cs [15,31,32]. Indeed, the Gibbs free energy of the adsorption of cations on clay minerals is very different among the several mineralogical species [24,33].

It has been suggested that the above-reported correlation could be expressed as a linear function of the SOM following the model:ln(V_E_) = ln(V_0_) − Kds_OM_ × SOM

Following the proposed linear model, ln(V_0_) is the logarithm of the downward migration rate in absence of organic matter, and Kd_SOM_ is the fraction of the mobile element retained in the layer because of the adsorption of the element into the organic matter. The values of ln(V_0_) and Kd_SOM_ have been estimated by linear regression above.

It is reasonable to assume both the parameters ln(V_0_) and Kd_SOM_ could be related to chemical properties of the elements.

With respect to anions, no meaningful correlation was found between element properties and ln(V_0_) or Kd_SOM_. With respect to bivalent cations Co, Ni, Cu, Zn, Mn, Cd, Ca and Pb, the parameter ln(V_0_) is linearly correlated to the ΔG* of the adsorption of the cation on bentonite for the divalent cations, as shown in the following equation:ln(V_0_) = (0.689 ± 0.128) + (0.0123 ± 0.0047) × ΔG*; n = 6, r^2^ = 0.634, F = 5.80 × 10^−2^

The downward migration rate (V_0_) of the element therefore decreases exponentially as the adsorption constant on bentonite increases. The parameter Kd_SOM_ for the bivalent cations Co, Ni, Cu, Zn, Mn, Cd, Ca and Pb is resulted inversely proportional to the ionic radius of the element following the linear model:

Kd_SOM_ = (0.026 ± 0.049) + (0.471 ± 0.121) × IP − 1; n = 8, r^2^ = 0.671, F = 7.93 × 10^−3^


Assuming that the soil properties that control the downward migration rates of the elements do not change with time and are in the steady state, the proposed box and flux model allowsus to predict the evolution with time of the element concentration throughout the soil profile. This assumption of the steady state of the soil properties controlling the PTEs leaching is reasonable when the concentration of the SOM pool in the soil profile has reached the steady state. It is not true if the climate change impact on land biomass and plant productivity are considered.

It is evident from the applied model that the time needed to decreases at half the concentration of the pollutants in the ploughed layer ranges from 25 years for the slowest mobile element (S) to 10 years for the faster moving (Na). The distribution of the pollutant throughout the soil profile calculated by the proposed box and flux model does not follow a Gaussian shape distribution, unlike in the case of the usual convective–advective physical models. It is because some soil reservoirs selectively accumulate some PTEs with respect to others PTEs. The reason for this phenomenon is that the proposed box and flux model is sensitive to the inhomogeneity commonly observed in the layered soil profile.

## 4. Conclusions

The main pedological processes that are active in the soil profile are ruled by SOM accumulation; weathering of the primary minerals, particularly the mica family; the formation of swelling clay minerals, which are stabilized by the very high soil base saturation; and the translocation and precipitation of soil colloids and oxide phases. The soil profile studied is characterized by almost constant pH values across the profile, which range from acid to slightly-acid. A sharp pH decrease is observed at the top of the Bt horizon, where it is reasonable to suppose it is buffered by iron oxide precipitation. As we observed soil structures and the related colors, wide Eh values are expected because of seasonal weather fluctuations and local soil structures at microscales. Among the elements studied K, Mg, Al and Si extracted by aqua-regia are predominantly derive from illitic minerals, while the aqua-regia extractable pool of Co, Ni, Cu, Zn, Cd, Pb, Mn and Cr is soil contamination derivedfrom the local smelter plant emissions tothe atmosphere and deposition on the topsoils. The soil-to-crust enrichment factor of the PTEs is explained by the element boiling point. As the boiling point decreases, the soil-to-rock enrichment factor increases, and with increasing boiling point, the SEF decreases. The same trends were observed in Northern Italy, the USA and in Canadian soils [13,15,16,18,31]. This demonstrates that extensive impact industrial emissions can have on food production. The average downward migration rates range from 0.89 to 1.75 cm^x^y^−1^, in the following order:S_(0.89)_ < Zn_(0.92)_ = P_(0.92)_ ≈ Pb_(0.93)_ < Mn_(1.06)_ < Cu_(1.14)_ < Ca_(1.21)_ ≈ Ni_(1.23)_ ≈ Co_(1.,24)_ ≈ Si_(1.26)_ < Cd_(1.42)_ < Na_(1.75)_

The main soil component retarding the leaching of the monovalent Na is the clay fraction. The SOM is not effective in Na adsorption in a detectable way. The main soil component adsorbing the divalent cations and anionic PTEs is the SOM. The SOM and, to the lesser effect, the soil organic Bentonite phase are the main soil components affecting the PTEs mobility. The affinity of the PTEs with the SOM is inversely correlated to the IP. The residence times of the individual PTEs in soil are in the following order:Na_(3.3)_ < Ca_(4.8)_ < Co_(5.5)_ ≈ Si_(5.7)_ < Ni_(6.3)_ ≈ Cu_(6.2)_ < Zn_(7.5)_ < P_(7.8)_ < S_(10.1)_ < Cd_(11.7)_ < Pb_(23.9)_ < Mn_(28.9)_

The residence time of the Na is ruled by adsorption on clay minerals. The residence time in soils of divalent cations Ca, Co, Ni, Cu, Zn, Cd, Pb and Mn, and the anions Cr, S, Si and P, are ruled by adsorption on SOM. The main PTE property which explains the PTE affinity to the SOM is the IP. The correlations found among the measured PTEs and affinity to soil matter should allow us to estimate the affinity to SOM of PTEsnot-measured in our study, such as W, Se, As or the Lanthanides elements for which few data are reported in the literature. The biogeochemical kinetics of the trace PTEs obey first-order kinetics in the box and flux model. The box and flux models proposed and tested here allow us to forecast the downward migration rates and the theoretical time to self-clean polluted soil. The box and flux models describing the leaching in soils of PTEs do not follow the Gaussian distribution shape typical of the physical advective–diffuse models because it is sensitive to layered soil horizon in homogeneity. The result obtained in the Scottish soil agree with the data obtained investigating Alpine soils developed on crystalline basement [16,23,34]. The box and flux model has provided a useful geochemical tool to investigate, describe and forecast the fate of trace PTEs contaminating the environmental ecosystems, as well as the potential human health impact of soil contamination by smelter plant activity [13].

## Figures and Tables

**Figure 1 ijerph-18-08930-f001:**
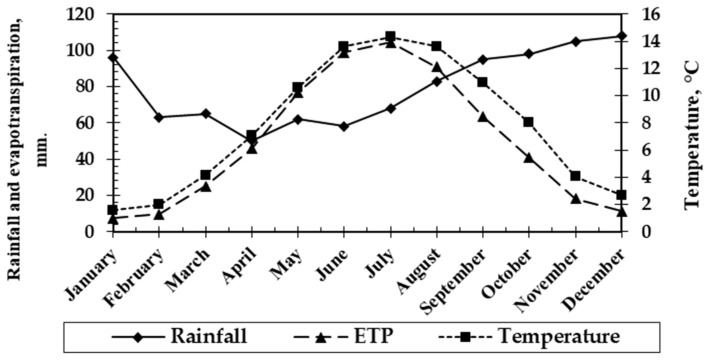
Regional climatic data influencing Clelland County, Glasgow, Scotland (UK).

**Figure 2 ijerph-18-08930-f002:**
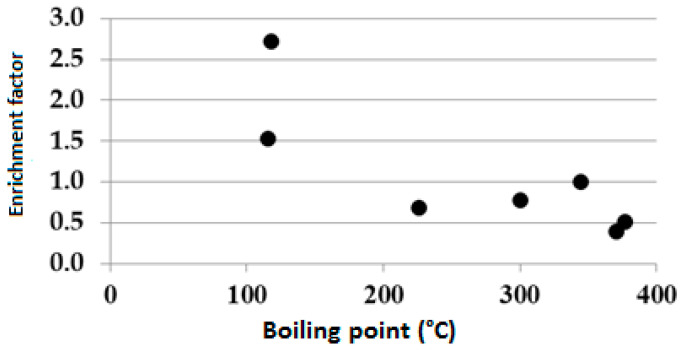
SEF versus boiling point at Clelland.

**Figure 3 ijerph-18-08930-f003:**
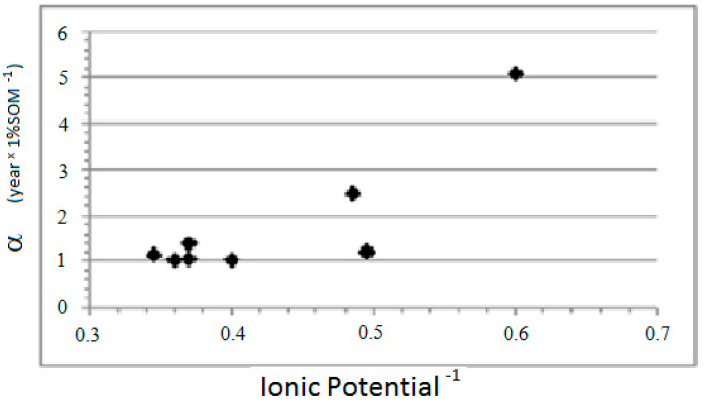
Values of the affinity to the SOM α versus IP^−1^ for the bivalent cations.

**Figure 4 ijerph-18-08930-f004:**
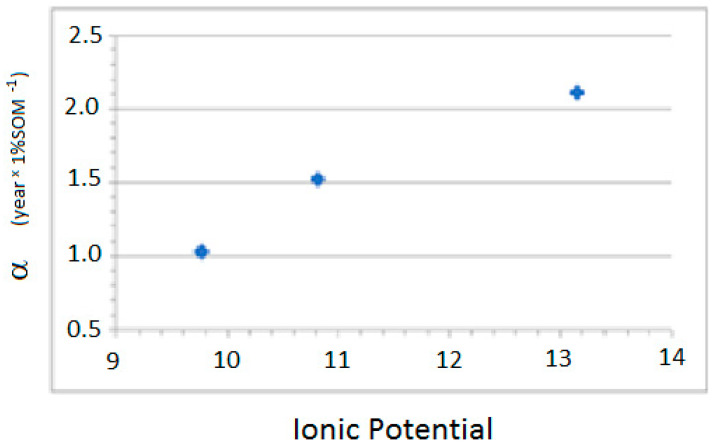
Values of the affinity to the SOM α versus IP for anions.

**Table 1 ijerph-18-08930-t001:** Soil properties.

Depth	Moisture	Sand	Clay	SOM	CEC	pH	pH	ΔpH
(cm)	%	%	%	%	Eq/kg	Water	KCl	
0–5	36.6	3.27	33.60	11.59	10.83	5.10	4.40	−0.70
5–10	32.8	10.23	20.00	7.98	9.82	4.80	4.41	−0.39
10–15	24.5	4.86	20.90	5.97	9.75	4.85	4.42	−0.43
15–20	21.3	6.74	20.00	5.00	9.77	4.88	4.39	−0.49
20–25	21.2	5.91	24.20	5.16	8.99	4.85	4.44	−0.41
25–30	15.7	4.22	24.10	2.98	7.48	4.78	4.39	−0.39
30–35	16.6	3.47	23.00	2.15	7.03	4.76	4.27	−0.49
35–40	21.1	2.66	31.10	1.90	8.18	4.71	4.03	−0.68
40–45	31.8	0.00	76.02	1.96	10.6	4.50	3.71	−0.79

**Table 2 ijerph-18-08930-t002:** Pseudo-total pool concentration (average values).

Depth	Al	Ca	Fe	K	Mg	Mn	Na	P	S	Si	Cd	Co	Cr	Cu	Ni	Pb	Zn
(cm)	ppm																
0–5	12,826	2050	25,701	1130	1425	557	30	382	529	289	2.03	9.22	192	35	34	42	108
5–10	13,487	2135	28,916	1093	1434	598	12	370	506	266	2.63	9.50	128	36	30	45	179
10–15	12,180	2395	30,045	934	1415	648	42	377	531	230	1.73	11.02	98,	60	30	42	128
15–20	15,142	2264	29,975	1240	1583	650	84	391	496	268	1.01	11.17	98	45	32	40	119
20–25	12,384	2097	29,200	981	1413	685	128	346	460	253	0.61	10.46	84	33	30	30	100
25–30	12,650	1747	29,104	1084	1446	864	100	325	271	260	0.25	11.42	74	28	27	24	111
30–35	15,402	1480	29,978	1454	1908	639	73	280	250	345	1.42	8.01	71	28	28	27	154
35–40	14,848	1768	29,350	1343	1622	722	126	338	323	305	2.47	8.22	68	46	26	40	63
40–45	23,219	2594	40,389	3760	4364	454	119	187	187	371	1.67	13.12	88	37	42	53	71

**Table 3 ijerph-18-08930-t003:** Correlation matrix among elements of pseudo-total pool (bold: *p* < 0.05).

	Al	Ca	Fe	K	Mg	Mn	Na	P	S	Si	Cd	Co	Cr	Cu	Ni	Pb	Zn
**Al**	1.000																
**Ca**	0.401	1.000															
**Fe**	**0.921**	0.536	1.000														
**K**	**0.979**	0.456	**0.936**	1.000													
**Mg**	**0.971**	0.473	**0.952**	**0.997**	1.000												
**Mn**	−0.627	−0.601	−0.507	−0.634	−0.634	1.000											
**Na**	0.384	−0.079	0.438	0.374	0.371	0.255	1.000										
**P**	**−0.854**	−0.094	**−0.854**	**−0.886**	**−0.897**	0.372	−0.493	1.000									
**S**	−0.575	0.392	−0.540	−0.565	−0.561	−0.142	−0.608	**0.826**	1.000								
**Si**	**0.838**	−0.090	0.629	**0.801**	0.783	−0.527	0.313	**−0.837**	−0.718	1.000							
**Cd**	0.137	0.120	−0.023	0.101	0.070	−0.508	−0.466	0.077	0.233	0.220	1.000						
**Co**	0.462	0.754	0.665	0.541	0.565	−0.197	0.253	−0.360	−0.073	−0.019	−0.434	1.000					
**Cr**	−0.227	0.253	−0.396	−0.168	−0.192	−0.443	−0.701	0.424	0.648	−0.164	0.384	−0.110	1.000				
**Cu**	0.075	0.518	0.053	−0.097	−0.085	−0.117	−0.135	0.341	0.466	−0.370	0.346	0.148	−0.015	1.000			
**Ni**	0.759	0.753	0.713	**0.812**	**0.814**	**−0.841**	−0.002	−0.531	−0.038	0.500	0.117	0.643	0.331	0.034	1.000		
**Pb**	0.561	0.780	0.501	0.568	0.551	−0.794	−0.229	−0.179	0.252	0.267	0.657	0.347	0.390	0.478	0.735	1.000	
**Zn**	−0.410	−0.199	−0.383	−0.449	−0.426	0.014	−0.763	0.359	0.344	−0.297	0.113	−0.295	0.224	−0.171	−0.268	−0.200	1.000

**Table 4 ijerph-18-08930-t004:** Soil enrichment factor (SEF) in USA and Clelland (average values).

Element	SEF USA *	SEF Clelland +	Boiling Point, K
**Co**	0.455	0.512	3200
**Ni**	0.238	0.389	3186
**Cr**	0.540	1.001	2945
**Cu**	0.500	0.773	2840
**Mn**	0.579	0.680	2235
**Pb**	1.357	2.716	2022
**Ca**	0.585	0.050	1757
**Zn**	0.800	1.531	1180
**Cd**	3.182	13.960	1040
**S**	6.154	1.542	718
**P**	0.430	0.333	550

* Bowen [31]; + present investigation.

**Table 5 ijerph-18-08930-t005:** Residence time (years) of PTEs in the soil reservoir (**** = not determinable).

Depth	Na	Ca	Co	Ni	Cu	Zn	Cd	Pb	Mn	Si	P	S
(cm)	Years											
0–5	6.4	16.,8	14.3	20.0	14.5	15.4	32.3	66.7	67.4	17.1	22.1	33.0
5–10	1.8	8.8	7.4	8.7	7.4	15.4	31.3	47.6	36.2	7.8	10.6	14.8
10–15	3.0	6.6	5.7	5.7	8.5	7.8	11.2	31.3	26.2	4.5	7.1	10.3
15–20	3.3	4.8	4.4	4.4	5.4	5.3	4.5	20.0	19.9	3.8	5.6	7.2
20–25	3.6	6.7	3.4	3.3	3.3	3.6	2.2	13.9	16.9	2.8	4.0	5.3
25–30	2.5	2.7	3.3	2.6	2.5	3.4	1.0	2.1	19.3	2.5	3.4	2.9
30–35	1.8	2.3	2.3	2,.7	2.3	6.0	3.3	1.5	15.3	3.2	3.1	2.7
35–40	4.4	3.1	2.8	2.9	5.4	3.4	7.6	7.9	29.9	3.6	6.8	5.4
40–45	****	****	****	****	****	****	****	****	****	****	****	****
Average	3.3	6.1	5.5	6.3	6.2	7.5	11.7	23.9	28.9	5.7	7.8	10.1
SD, %	45.3	73.4	72.4	94.0	65.5	67.4	109.5	97.5	59.2	86.7	79.9	98.5

**Table 6 ijerph-18-08930-t006:** Correlation parameter between residence time and organic matter.

Element	A	SD%	r^2^	F
**Pb**	5.08	10.1	0.933	6.10 × 10^−5^
**Ca**	1.21	7.7	0.960	1.31 × 10^−3^
**Cd**	2.47	17.3	0.828	1.15 × 10^−4^
**Mn**	1.03	11.0	0.923	5.00 × 10^−6^
**Co**	1.05	6.7	0.967	7.01 × 10^−6^
**Zn**	1.40	10.4	0.929	7.44 × 10^−5^
**Ni**	1.03	14.8	0.912	1.42 × 10^−4^
**Cu**	1.14	9.6	0.938	4.81 × 10^−5^
**Si**	1.13	15.1	0.914	1.52 × 10^−4^
**P**	1.52	11.2	0.919	1.12 × 10^−4^
**S**	2.11	11.6	0.914	1.32 × 10^−5^

**Table 7 ijerph-18-08930-t007:** Downward migration rates of the pollutants, cm·year^−1^ (*** = not determinable).

Sample	V_S_	V_Zn_	V_P_	V_Pb_	V_Mn_	V_Cu_	V_ca_	V_Ni_	V_co_	V_Si_	V_Cd_	V_Na_
**Fapa1.1/9**	0.16	0.33	0.23	0.08	0.37	0.35	0.30	0.25	0.35	0.29	0.16	0.79
**Fapa1.2/9**	0.34	0.33	0.47	0.11	0.69	0.68	0.57	0.58	0.68	0.64	0.16	2.80
**Fapa1.3/9**	0.48	0.64	0.70	0.16	0.96	0.59	0.75	0.88	0.88	1.11	0.45	1.69
**Fapa1.4/9**	0.69	0.94	0.90	0.25	1.25	0.93	1.04	1.13	1.14	1.32	1.12	1.52
**Fapa1.5/9**	0.95	1.41	1.26	0.36	1.48	1.52	1.37	1.53	1.48	1.76	2.30	1.40
**Fapa1.6/9**	1.75	1.46	1.47	2.40	1.30	2.02	1.84	1.89	1.51	2.00	5.00	1.98
**Fapa1.7/9**	1.85	0.83	1.62	3.45	1.63	2.14	2.23	1.85	2.14	1.54	1.51	2.72
**Fapa1.8/9**	0.93	1.46	0.73	0.63	0.84	0.93	1.59	1.74	1.78	1.40	0.66	1.14
**Fapa1.9/9**	***	***	***	***	***	***	***	***	***	***	***	***
**Average**	0.89	0.92	0.92	0.93	1.06	1.14	1.21	1.23	1.24	1.26	1.42	1.75
**SD, %**	69.63	51.99	52.90	137.31	40.01	58.65	54.81	50.37	47.87	44.96	114.29	40.81

**Table 8 ijerph-18-08930-t008:** Correlation between ln(V_E_) and soil properties (bold: *p* < 0.01).

ln V_E_	Moisture	Coarse Fraction	Clay	SOM	CEC	pH	pH
	%	%	%	%	Eq/kg	water	KCl
**ln V_Co_**	**−0.935**	−0.254	−0.265	**−0.975**	**−0.887**	−0.837	−0.474
**ln V_Ni_**	**−0.961**	−0.218	−0.313	**−0.971**	**−0.864**	−0.835	−0.394
**ln V_Cu_**	**−0.894**	−0.122	−0.343	**−0.822**	**−0.898**	−0.682	−0.145
**ln V_Zn_**	**−0.877**	−0.395	−0.024	**−0.836**	−0.683	−0.590	−0.384
**ln V_Cd_**	**−0.914**	−0.276	−0.,235	**−0.743**	**−0.742**	−0.470	−0.024
**ln V_Pb_**	**−0.882**	−0.440	−0.094	**−0.848**	**−0.977**	−0.647	−0.376
**ln V_Mn_**	**−0.904**	0.005	−0.594	**−0.803**	**−0.715**	−0.650	−0.008
**ln V_Ca_**	**−0.962**	−0.282	−0.253	**−0.968**	**−0.928**	−0.808	−0.421
**ln V_Na_**	−0.367	0.509	−0.790	−0.399	−0.488	−0.623	0.156
**ln V_Si_**	**−0.954**	−0.121	−0.431	**−0.909**	**−0.771**	−0.769	−0.211
**ln V_P_**	**−0.954**	−0.111	−0.481	**−0.865**	**−0.834**	−0.704	−0.100
**ln V_S_**	**−0.971**	−0.260	−0.303	**−0.938**	**−0.937**	−0.770	−0.315

**Table 9 ijerph-18-08930-t009:** Parameters of the regression between ln(V_E_) and SOM.

Element	B	SD	C	SD	r^2^	F
**Pb**	1.018	0.571	0.365	0.093	0.720	3.48 × 10^−3^
**Ca**	1.093	0.130	0.120	0.021	0.938	7.73 × 10^−5^
**Cd**	1.259	0.637	0.281	0.104	0.698	3.48 × 10^−2^
**Mn**	0.615	0.224	0.120	0.036	0.645	1.64 × 10^−2^
**Zn**	0.625	0.265	0.160	0.043	0.698	9.76 × 10^−3^
**Cu**	0.824	0.,278	0.160	0.045	0.676	1.22 × 10^−2^
**Ni**	1.164	0.130	0.211	0.021	0.944	5.69 × 10^−5^
**Co**	1.033	0.101	0.177	0.016	0.951	3.10 × 10^−5^
**S**	0.,911	0.223	0.240	0.036	0.880	5.70 × 10^−4^
**P**	0.687	0.252	0.173	0.041	0.748	5.53 × 10^−3^

## Data Availability

Data available from first author.

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
