# Peer review of "Development and Validation of a Box and Flux Model to Describe Major, Trace and Potentially Toxic Elements (PTEs) in Scottish Soils"

_ijerph, 2021, doi:10.3390/ijerph18178930_

Round 1

Reviewer 1 Report

General comments:

What I missed is a proper definition of PTEs because some people will not consider P, S, Zn as PTEs. On what basis did you classify the elements determined here as PTEs? Except I missed, I didn’t see the results of your model validation or even model validation parameters. Moreover, there was no attempt to properly discuss the result as the results and discussion section was largely a mere listing of the results.

Specific comments:

Lines 79-81: here you have regional monthly data. What year is this for?

Lines 83-97: Did you study the profile or this information was gleaned from another study? Make it clear

Lines 112-113: the sentence is not clear

Lines 132-137: what was your justification for modifying the Bonazzola et al. model?

Line 299: The presentation of Table 3 is poor and clumsy making it difficult to read

Also, the result and discussion was a mere listing of the results without a proper discusion of the research.

Author Response

  • We accepted the referee's suggestions, and we hope to have improved the paper. The title and abstract have been revised and the introduction rewritten. Furthermore, the discussion of the results was integrated.
  • Line 79-81: we hope to have clarified the point
  • Line 83-97: we hope to have clarified that the profile is the one on which the model was applied
  • Line 132-133: we hope to have well justified the changes to the Bonazzola et al.  model
  • Line 299: Table 3 is table 3 has been made readable
  • The conclusion section has been modified.

Reviewer 2 Report

In this manuscript authors have developed and validated a box and flux model describing the behavior of PTEs in a soil from Central Scotland polluted by historical atmospheric emissions from a smelter plant. This a pure academic exercise with no or little relevance to the impacted public as they are treating the soil as a dead material. Natural soils are a living system and if the study to be meaningful they should include the interaction of microbes, plants and roots with trace elements.  Hence, this can only be accepted as a technical note to inform the public about an in-depth future study. For a technical note, authors should condense the manuscript and resubmit for re-review.

Author Response

  • In our opinion this is not a technical note or even a pure academic exercise. It is well known that soil is not dead material and that the biotic component plays an important role in the cycle of elements but has not been deliberately considered. It is a good suggestion for in-depth work.
  • We accepted the referee's suggestions, and we hope to have improved the paper. The abstract have been revised and the introduction rewritten. Furthermore, the discussion of the results was integrated. The conclusion section has been modified.

Reviewer 3 Report

The authors developed a box and flux model to describe trace element mobility in soils. Overall, the manuscript was well written. However, I have some concerns that should be addressed. I recommend its acceptance after minor revision. Title:Are the elements studied all trace elements, such as S, Ca, P and Na. If not, please revise the title. L52-68: The authors introduced many models, but did not clarify whether there are some models to describe PTE mobility in soils. In another word, what about the novelty of the present study? L78: revised as “an adjacent decommissioned smelter”, “XX” as “20th” L185-192: References are needed; please also check the whole manuscript; Table 2, Table 4: Are the values in these tables average values? What about SE or SD? Table 3, Table 6: How to analyze Correlation? Pease indicate in 2.7.

Author Response

We accepted the referee's suggestions, and we hope to have improved the paper. The abstract have been revised and the introduction rewritten.